# Identification of Benzophenone Analogs in Rice Cereal through Fast Pesticide Extraction and Ultrahigh-Performance Liquid Chromatography–Tandem Mass Spectrometry

**DOI:** 10.3390/foods11040572

**Published:** 2022-02-16

**Authors:** Xuan-Rui Liu, Yu-Fang Huang, Jun-Jie Huang

**Affiliations:** 1Department of Safety, Health and Environmental Engineering, National United University, Miaoli 36063, Taiwan; aass1aass35@gmail.com (X.-R.L.); k10220202@gmail.com (J.-J.H.); 2Center for Chemical Hazards and Environmental Health Risk Research, National United University, Miaoli 36063, Taiwan; 3Institute of Food Safety and Health Risk Assessment, National Yang Ming Chiao Tung University, Taipei 11221, Taiwan; 4Institute of Environmental and Occupational Health Sciences, National Yang Ming Chiao Tung University, Taipei 11221, Taiwan

**Keywords:** Benzophenone analogs, rice cereal, FaPEx, ultrahigh–performance liquid chromatography–tandem mass spectrometry

## Abstract

A fast, robust, and sensitive analytical method was developed and validated for the simultaneous identification of benzophenone (BP) and nine BP analogs (BP-1, BP-2, BP-3, BP-8, 2-hydroxybenzophenone, 4-hydroxybenzophenone, 4-methylbenzophenone [4-MBP], methyl-2-benzoylbenzoate, and 4-benzoylbiphenyl) in 25 samples of rice cereal. Fast pesticide extraction (FaPEx) coupled with ultrahigh-performance liquid chromatography–tandem mass spectrometry was applied. The developed method exhibited satisfactory linearity (r > 0.997), favorable recoveries between 71% and 119%, and a limit of detection ranging from 0.001 to 0.5 ng/g. The detection frequencies of BP, 4-MBP, and BP-3 were 100%, 88%, and 52%, respectively. BP had higher geometric levels, with a mean of 39.8 (19.1–108.9) ng/g, and 4-MBP had low levels, with a mean of 1.9 (1.3–3.3) ng/g. The method can be applied to routine rice cereal analysis at the nanogram-per-gram level. For infants aged 0–3 years, the hazard quotients of BP and 4-MBP were lower than one, and the margin of exposure for BP was higher than 10,000, suggesting that rice cereal consumption poses no health concern for Taiwanese infants.

## 1. Introduction

Rice cereal is an essential product for infant nutrition in Asia and plays a key dietary role. It consists of approximately 75% carbohydrates and 7–9% protein, and a very small amount of fiber and minerals [1]. The demand for cereal-based products has increased, with yearly global cereal production reaching 2.791 billion tons [2]. Benzophenone (BP) and BP analogs (BPs) are a part of a group of endocrine-disrupting chemicals that can interfere with hormone action [3]. On the basis of a report by the National Toxicology Program on carcinogenicity in mice and rats with carcinoma, hepatoblastoma, and hepatocellular and renal tubule adenomas, the International Agency for Research on Cancer classified BP as possibly carcinogenic for humans (Group 2B) [4,5,6].

BPs are used as ultraviolet (UV) filters in personal care products, as UV stabilizers, and as UV-curable inks in printed food contact materials (FCM) [4,7]. Compounds such as BP, 4-methylbenzophenone (4-MBP), 4-benzoylbiphenyl (PBZ), and methyl-2-benzoylbenzoate (M2BB) have a strong tendency to migrate into food and cause contamination [8]. In 2009, the Rapid Alert System for Food and Feed announced that as much as 798 ng/g of 4-MBP can migrate from cardboard packaging materials into muesli [9]. The European Union has regulated a specific migration limit of 600 μg/kg for BP in food [10]. BPs have been reported to occur as photoinitiators [11] and occur naturally in foods such as grapes and black tea and are added to foods as flavoring agents [4,12]. Humans are mainly exposed to BPs through inhalation [13], dermal absorption [14], and ingestion [4]. Studies have detected BPs in human fluids such as urine, blood, and breast milk [15].

Diet is a major source of BP exposure in the general population. Humans exposure to BPs from FCM occurs as a result of migration from materials into foodstuffs and foods. Although BP levels in foodstuffs are too low to constitute a risk, long-term and cumulative exposure to BPs may have adverse effects. Thus, a convenient and reliable analytical method for quantifying BP contents in foods is required. Pollutant extraction is a critical point in chemical analyses [16]. Solvent extraction with acetonitrile [9,17] and pressurized liquid extraction (PLE) [18] have been proposed for determining the level of BPs in breakfast cereals; however, these techniques are time-consuming and require a large volume of organic solvents. Solvent extraction followed by solid-phase extraction (SPE) with a hydrophilic–lipophilic balance (HLB) or silica cartridge [19,20,21]; the quick, easy, cheap, effective, rugged, and safe (QuEChERS) method [22]; and QuEChERS combined with SPE have been proposed [23]. However, some methods are limited in terms of instrument sensitivity and number of BPs that can be simultaneously analyzed. Fast pesticide extraction (FaPEx) is a simplified version of the QuEChERS method and is based on the same principles. It involves using single-use prefilled sealed cartridges [24]. FaPEx is simple and fast and can be used to extract pesticide residues in agricultural samples. Chromatographic methods combined with MS represent a predominant determination practice. Gas chromatography (GC)–mass spectrometry (MS) has been frequently applied to the analysis of some FCM substances and BPs in foods [9,11,18,19]. BP content in cereal-based products is also quantified through high-performance liquid chromatography (HPLC) with UV detection [25], HPLC combined with diode array detection [17,26], HPLC combined with MS [9], and tandem MS (MS/MS) [17,22,23,27].

This study developed a FaPEx technique coupled with ultrahigh-performance liquid chromatography (UHPLC)–MS/MS to simultaneously analyze the concentrations of BP and nine BP analogs (BP-1, BP-2, BP-3, BP-8, 2-hydroxybenzophenone (2-OHBP), 4-hydroxybenzophenone (4-OHBP), 4-MBP, M2BB, and PBZ). The effectiveness of the method was demonstrated through analysis of commercial samples of rice cereal in Taiwan. To ensure accurate quantification and reduce measurement uncertainty, the isotope-labeled internal standards (ILISs) for UHPLC–MS/MS were used.

## 2. Experiment

### 2.1. Reagents and Chemicals

The analytical standards of BP, 2-hydroxybenzophenone (2-OHBP), 4-hydroxybenzophenone (4-OHBP), methyl-2-benzoylbenzoate (M2BB), and 4-methylbenzophenone (4-MBP) were purchased from Sigma-Aldrich (St. Louis, MO, USA). In addition, 2,4-dihydroxybenzophenone (BP-1) and 2-hydroxy-4-methoxybenzophenone (BP-3) were purchased from AccuStandard (New Haven, CT, USA), 2,2′, 4,4′-tetrahydroxybenzophenone (BP-2) and 2,2′-dihydroxy-4-methoxybenzophenone (BP-8) were purchased from Tokyo Chemical Industry (Tokyo, Japan), and PBZ was purchased from Alfa Aesar (Lancashire, UK). The ILISs, d_5_-BP-1, d_3_-4-MBP, d_3_-BP-8, ^13^C_6_-diOHBP, d_4_-4-OHBP, and d_4_-BP-2 were purchased from Toronto Research (North York, Toronto, Canada); moreover, d_5_-BP, d_5_-BP-3, and the SPE bulk sorbent, primary secondary amine (PSA), were obtained from Sigma-Aldrich (Burlington, MA, USA). All standards and ILISs had purities of >97%. Anhydrous magnesium sulfate (MgSO4, 99% purity), liquid chromatography (LC)-grade acetonitrile (ACN), formic acid (88% purity), acetic acid (99.7% purity), and LC–MS-grade methanol (MeOH) were obtained from J.T. Baker (Phillipsburg, NJ, USA). FaPEx-cer was obtained from Silicycle (Quebec, Canada). Sodium chloride (NaCl) (>99% purity) was purchased from PanReac (Castellar del Vallès, Barcelona, Spain). Sepra C18-E (50 μm, 65 Å) was obtained from Phenomenex (Torrance, CA, USA).

### 2.2. Standard

A stock standard solution of each analyte (1000 mg/L) was individually prepared in ACN. To create a working solution in the range of 0.4–100 ng/g, a suitable volume of the solution was diluted with MeOH. BP-d_5_, d_5_-BP-1, d_3_-4-MBP, d_3_-BP8, ^13^C_6_-diOHBP, d_4_-4-OHBP, and d_5_-BP-3 were used as ILISs for BP, BP-1, 4-MBP, BP-8, 2-OHBP, 4-OHBP, and BP-3, respectively. Because of a lack of ILISs for BP-2, M2BB, and PBZ, d_4_-4-OHBP was used as an internal standard (IS) for BP-2 and M2BB, and d_5_-BP-3 was used as an IS for PBZ. The mixture of ILISs was prepared at 100 mg/L and diluted to 20 μg/L. All stock and working solutions of standards and ILISs were stored at −20 °C.

### 2.3. Sample Treatment

Two procedures, namely QuEChERS and FaPEx, were compared to determine which sample pretreatment method was suitable for rice cereal.

Method A: The QuEChERS procedure includes two steps: extraction by using ACN and salts for enhanced efficiency and cleanup by using dispersive SPE. In these steps, sorbents such as MgSO4, PSA, and C18 are used to remove excess water, pigments, and lipids and fatty acids, respectively. The QuEChERS method employed in this study has been described previously [28]. A homogenized rice cereal sample (5 g) was placed in a 50 mL polypropylene (PP) tube, and an ILIS (8 ng/g), deionized water (10 mL), and ACN (10 mL) with 1% acetic acid were added. The mixtures were shaken for 1 min, extraction salt packages (4 g of anhydrous MgSO_4_ and 1 g of NaCl) were added, and the mixtures were shaken again vigorously for 1 min and centrifuged for 5 min at 6000× *g*. The supernatant was added to the cleanup sorbents containing 1.2 g of MgSO4, 1.2 g of C18-E, and 0.4 g of PSA. Finally, the residue was reconstituted in 200 μL of MeOH and filtered using 0.22 μm polytetrafluoroethylene filters.

Method B: The FaPEx method is simple and involves using single-use prefilled sealed cartridges with MgSO_4_, PSA, C18, and graphitized carbon black. For FaPEx, a homogenized cereal sample (0.5 g) was loaded into a 15 mL PP tube. Subsequently, pure water (1 mL) was added, and the sample was spiked with an ILIS (8 ng/g) and vortexed for 1 min. After the sample was allowed to stand for 30 min, 5 mL of ACN with 1% acetic acid was added; the mixtures were vortexed for 30 s and centrifuged at 6000× *g* for 5 min. Thereafter, the supernatant solution was transferred to a FaPEx-cer kit, with the liquid flow rate controlled at 1 drop/s and the dryness level controlled using a gentle nitrogen stream. Finally, the residue was treated in the same manner as the QuEChERS procedure. Figure 1 presents a flowchart of sample preparation.

### 2.4. Instrumentation

The BP and BP analogs, except for BP-2, were detected using a Nexera UHPLC–MS/MS system (Shimadzu, Kyoto, Japan) connected to a triple-quadrupole MS system (Shimadzu 8045, Kyoto, Japan) with an electrospray ionization (ESI) positive mode. BP-2 was detected in ESI negative mode during 0–6 min and switched to positive mode for 6.1–13.5 min during the run. The analytical column was a UPLC Waters BEH C18 column (Milford, MA, USA; 1.7 μm, 2.1 mm × 100 mm). The flow rate of the mobile phases, LC–MS-grade methanol containing 0.1% formic acid (A), and deionized water (B), was 0.3 mL/min with a gradient of 20–80% A at 3.5 min, 80% A at 1 min, 80–90% A at 1 min, 90% A at 4 min, and 90–20% A at 0.1 min; the gradient was re-equilibrated at 20% for 3.9 min. The total analysis time and sample injection volume were 13.5 min and 10 µL, respectively. BPs were monitored under multiple reaction monitoring modes. The flow rates of nebulizing gas, heating gas, and drying gas were 3, 10, and 10 L/min, respectively. The interface temperature, desolvation line temperature, and heat block temperatures were 300, 240, and 400 °C. LabSolution (version 5.93, Shimadzu, Kyoto, Japan) was used for data analysis. Table 1 presents the MS parameters, ion transitions for quantification and qualifications, retention time, and collision energy; Appendix A lists the respective mass spectra of the BP and nine BP analogs.

### 2.5. Method Validation Procedure and Real Sample Analysis

The method was validated in accordance with guidelines established in the United States [26] in terms of linearity, limit of detection (LOD), limit of quantification (LOQ), matrix effect, precision, and recovery. Matrix samples with a tin can or laminated aluminum foil packaging were selected as blank samples because they contained low BP levels (signal to noise ratio <10). Linearity was evaluated using solvent-matched and matrix-matched calibration standards covering seven levels (0.4, 2, 4, 8, 12, 20, and 100 ng/g, with an ILIS of 8 ng/g). Calibration curves of the solvents and matrix were obtained by plotting the quotients of the peak areas of BP and the nine BP analogs and their corresponding ILISs versus the levels of the standards. The matrix effect was evaluated through comparison of the slopes of the standards in a solvent with matrix-matched standards. LOD and LOQ were defined as the levels with signal to noise ratios of 3 and 10, respectively. Blank rice cereal samples with a 20 ng/g spiking level were used to evaluate the precision and accuracy of the method. Intraday and interday precision was used to determine the relative standard deviation (RSD). Intraday precision (*n* = 3) was assessed on the basis of the standard deviation (SD) of the recovery percentage of the spiked samples on a given day. Interday precision (*n* = 9) was determined by comparing the spiked samples across 3 days. Accuracy was evaluated on the basis of the mean recovery for these spiked samples (*n* = 9).

A total of 25 samples of rice cereal were purchased from supermarkets in Taiwan. Packaging materials and organic and nonorganic food were selected with consideration of proportionality. All samples were domestic. A total of 5, 12, and 8 samples were packaged in tin cans, laminated aluminum foil bags, and plastic bags, respectively, and 11 and 14 samples were organic and nonorganic, respectively.

### 2.6. Statistical Analysis

The experimental results are presented as means (SDs, range). The nonparametric Mann–Whitney U test and Kruskal–Wallis test were used to evaluate the differences in BP and BP analog levels between organic and nonorganic foods and among the packaging materials. Statistical analysis was performed using SPSS (version 19.0, SPSS Inc., Chicago, IL, USA), and the significance level was *p* < 0.05.

## 3. Results and Discussion

### 3.1. Extraction and Cleanup Method Selection

A comparison of the two methods for the extraction and cleanup of the BP and BP analogs revealed that FaPEx yielded higher recovery (Figure 2). FaPEx was established by the Taiwan Agricultural Chemicals and Toxic Substances Research Institute and is authorized for sale by Uni-Onward. FaPEx is more a convenient approach than QuEChERS, with an extraction time shorter than 5 min and low waste solvent production. In addition, FaPEx reduces the operating time and eliminates requirements for glassware and special equipment. Figure 3 presents the representative chromatogram of a rice cereal sample spiked with 20 ng/g of the BP and BP analog standards and 8 ng/g of the ILIS.

### 3.2. Method Validation

According to the solvent-matched and matrix-matched calibration curves, all analytes exhibited high linearity (r > 0.997), in the range of 0.4–100 ng/g (Table 2). The matrix effect of the BP and nine BP analogs was 64–210%, indicating that the matrix effect was present for BP-2, 2-OHBP, M2BB, and PBZ. Consequently, the matrix-matched standard solutions were selected for calibration. The LOD and LOQ were 0.001–0.512 and 0.003–1.536 ng/g, respectively. Accuracy was based on the calculation of the relative mean recovery in nine replicates, with a recovery range of 71–119%. Precision was expressed as the RSDs, and for intraday precision, three replicates of samples were analyzed during the day, while for interday precision, nine replicates of samples were assessed on three consecutive days. The intraday and interday RSDs range was 1.2–12.4% and 2.6–16.8%, respectively. Because of the simplicity of FaPEx, the sample was directly treated with a FaPEx cartridge after homogenization, and the resulting cleaned filtrate was ready for injection and chromatographic analysis. Therefore, FaPEx not only minimizes handling errors but also yields high recovery. The BP and BP analog analytes satisfied the validation criteria of the Codex Alimentarius of the United States [29].

To the best of our knowledge, this is the first study on the occurrence of BP and nine BP analogs in samples of rice cereal from Taiwan. GC–MS analysis is one of the frequently used techniques for the identification of compounds added intentionally and non-intentionally in plastic FCM because of its accessibility and sensitivity [11,30]. Tsochatzis et al. (2020) developed a liquid-liquid extraction technique with dichloromethane and 10% NaCl and GC–MS for quantification of 84 substances from plastic FCM [11]. Bugey et al. (2013) developed a technique involving PLE and GC–MS, and the LOQ of BP was 60 ng/g in cereal-based foodstuffs [18]. The European Food Safety Authority (EFSA; 2009) reported a pretreatment method involving solvent extraction with ACN, and analysis was performed through GC–MS or HPLC–MS, which yielded an LOQ of 50 ng/g for 4-MBP in breakfast cereals [9]. Hoeck et al. (2010) developed a solvent extraction method involving dichloromethane and ACN followed by SPE with a silica cartridge and performed GC–MS on breakfast cereals [19]. The LOD of BP was 2 ng/g, with a recovery of 74%. Jung et al. (2013) reported a solvent extraction method with ACN and HPLC–MS/MS for quantification of six photoinitiators in packaged foods [17]. The LODs were 38 and 2.5 ng/g for BP and 4-MBP, respectively, with recoveries of 89% and 94%, respectively. Chang et al. (2019) developed a QuEChERS method without a cleanup procedure and used UPLC–MS/MS to analyze 30 photoinitiators in breakfast cereals [22]. The LOQs were 20 and 10 ng/g for BP and 4-MBP, respectively, with recoveries of 62% and 120%, respectively. Gallart-Ayala et al. (2011) developed a QuEChERS and SPE method involving the use of an HLB cartridge and used HPLC–MS/MS to analyze 11 photoinitiators in packaged food [23]. The LOQs of BP and PBZ were 2.3 and 0.7 ng/g, respectively, with recoveries of 88% and 97%, respectively. Van den Houwe et al. (2016) developed a solvent extraction method with ACN, SPE, and an HLB cartridge and used UPLC–MS/MS to identify 17 photoinitiators in dry foodstuffs [27]. The LODs of BP and 4-MBP were 0.3 and 0.1 ng/g, respectively, with recoveries of 95% and 96%, respectively. These pretreatment techniques require a large amount of organic solvents and are time-consuming. This study developed a simpler, faster, and more efficient method with satisfactory results, lower LODs, and higher precision for the simultaneous identification of BP and nine BP analogs in rice cereal. The use of ILISs can ensure highly accurate quantification by reducing measurement uncertainty and increasing recovery.

### 3.3. Sample Analysis

The method was applied for BP and BP analog analysis in 25 Taiwanese rice cereals. Table 3 presents the BP and BP analog levels for the samples of rice cereals grouped by packaging material and chemical nature. Of the 10 analytes, three were detected in the range of 52–100% in the rice cereal; exceptions were BP-1, BP-2, BP-8, 2-OHBP, and M2BB, which were detected at percentages lower than the LOD. BP and 4-MBP were detected in 100% and 88% of the samples, and BP-3 was detected in >52% of the samples. These results suggest that BP and 4-MBP are prevalent in rice cereals, which is consistent with results reported in Switzerland [18], Belgium [9], Germany [17], and Spain [23]. In addition, PBZ and 4-OHBP were detected in 4% and 8% of the samples, respectively. The BP contributed the most to the total level in rice cereals, with a level of 45.8 ± 26.3 (19.1–108.9) ng/g. The second highest contributor was 4-OHBP, with a level of 23.6 ± 13.6 (<LOD–33.3) ng/g. In addition, 4-MBP and BP-3 had low levels: 1.9 ± 0.4 (<LOD–3.3) and 0.6 ± 0.1 (<LOD–0.7) ng/g, respectively. The range of BP levels was wider than those reported in cereal samples in Belgium [27] and Spain [23], which were <LOD–20 ng/g and 29–40 ng/g, but lower than those reported in Germany [17] (3367–3413 ng/g) and Switzerland [18] (5–7 × 106 ng/g). Levels of 4-MBP higher than those observed in this study have been reported in Belgium [9] and Germany [17]: 795–5400 and 65–8073 ng/g, respectively. The high levels of BP and 4-MBP in cereals may be attributable to food contamination from packaging or the packaging material in printed cardboard with additional plastic wrapping. Figure 4 displays the BP, 4-MBP, and BP-3 levels in the rice cereals by chemical nature and packaging material. No significant difference in mean BP, 4-MBP, and BP-3 levels was observed among the packaging materials or between the organic or nonorganic samples (all *p*-values > 0.05). Further studies with a large sample size are warranted to elucidate the relation between BPs and packaging material and chemical nature.

### 3.4. Risk Characterization

Because of the carcinogenicity and widespread prevalence of BP and 4-MBP in rice cereal, this study determined noncancer risk by using hazard quotients (HQs) and cancer risk using a margin of exposure (MOE) [31,32]. The HQ is the ratio of the estimated daily intake (EDI, ng/kg/day) to the chronic oral reference dose (RfD, ng/kg/day; HQ = EDI/RfD). In this study, EDI was estimated on the basis of the measured BP and 4-MBP levels (C, ng/g in weight), the daily ingestion rate (IR, g/day) of rice cereal, and the mean body weight (BW, kg) of infants (0–3 years of age) as follows: EDI = (C× IR)/BW. The maximum levels of BP and 4-MBP (109 and 3 ng/g, respectively) in the rice cereal was used to calculate HQ_worst_ (=EDI_maximum_/RfD). According to the Taiwan National Food Consumption Database [33] (2019; http://tnfcds.cmu.edu.tw/ accessed on 22 August 2020), the mean BW of infants (0–3 years of age) is 13 kg, and infants have the highest daily intake of rice cereal (21 g/day) among Taiwanese. The RfDs of BP and 4-MBP are 30 µg/kg/day [10]. In the worst-case exposure scenario of BP and 4-MBP, the HQ_worst_ poses a low risk, with values of 0.06 and 0.0002 for rice cereal consumption, respectively. In addition, the EFSA proposed a MOE metric for cancer risk assessment [31]. The MOE is the ratio of the lower bound of a 95% confidence interval on the benchmark dose (BMD) corresponding to a 10% higher number of tumor-bearing animals compared with control animals (BMDL_10_, mg/kg/day) to the EDI (MOE = BMDL_10_/EDI). EFSA (2009) reported that the BMDL_10_ for BP is 3.1 mg/kg/day [6]. The maximum EDI of BP (176 ng/kg/day) for infants (age 0–3 years) was used to calculate the MOE, and the MOE was 17,614. Generally, the higher MOE values represent lower health risk, and the value of carcinogenic risk accepted by the EFSA is 10,000 [34]. For infants aged 0–3 years, the HQs of BP and 4-MBP were lower than one, and the MOE for BP was higher than 10,000, indicating that rice cereal consumption poses no health concern for Taiwanese infants.

## 4. Conclusions

We developed a fast, simple, and robust FaPEx technique involving UHPLC–MS/MS to simultaneously analyze BP and BP analogs in samples of rice cereal. The method exhibited satisfactory results, with high recovery (from 71 to 119%) and LODs below the nanogram-per-gram (from 0.001 to 0.5 ng/g) level and precision (RSD less 17%) for BP and BP analogs analysis. BP and 4-MBP were the most abundant substances in the analyzed samples. The trace levels of BP-3 and 4-MBP observed in the samples indicate the necessity of developing analytical methods with high specificity and sensitivity; the proposed method satisfies these requirements. Simultaneous determination of BP and BP analogs in rice cereal makes it possible to evaluate human exposure to BPs. The HQs of two BPs were less than one, and the MOE of BP was deemed acceptable, indicating that exposure to BPs is unlikely to pose noncancer and cancer risks to Taiwanese infants.

## Figures and Tables

**Figure 1 foods-11-00572-f001:**
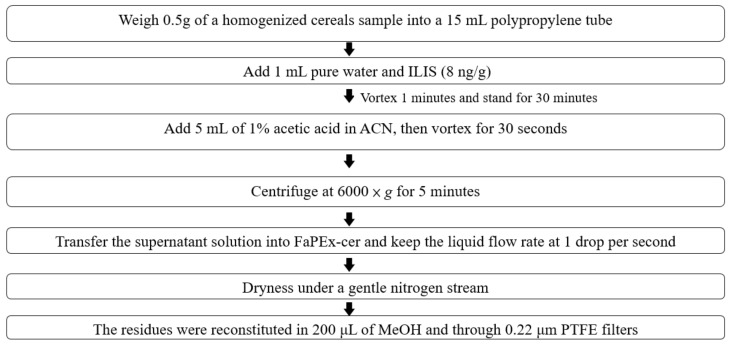
The flowchart of sample preparation: FaPEx.

**Figure 2 foods-11-00572-f002:**
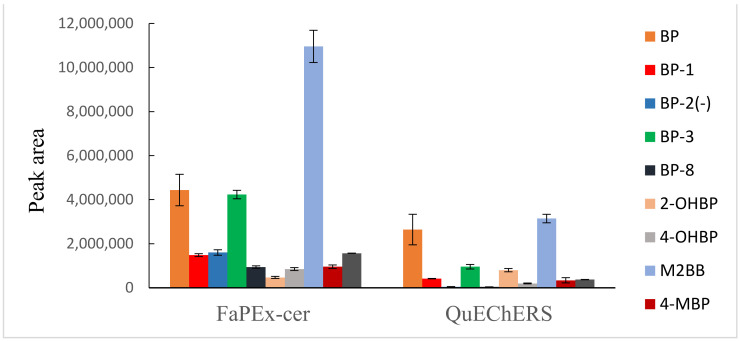
Comparison of FaPEx-cer and QuEChERs methods in rice cereal spiked with standards and ILISs (20 ng/g and 8 ng/g).

**Figure 3 foods-11-00572-f003:**
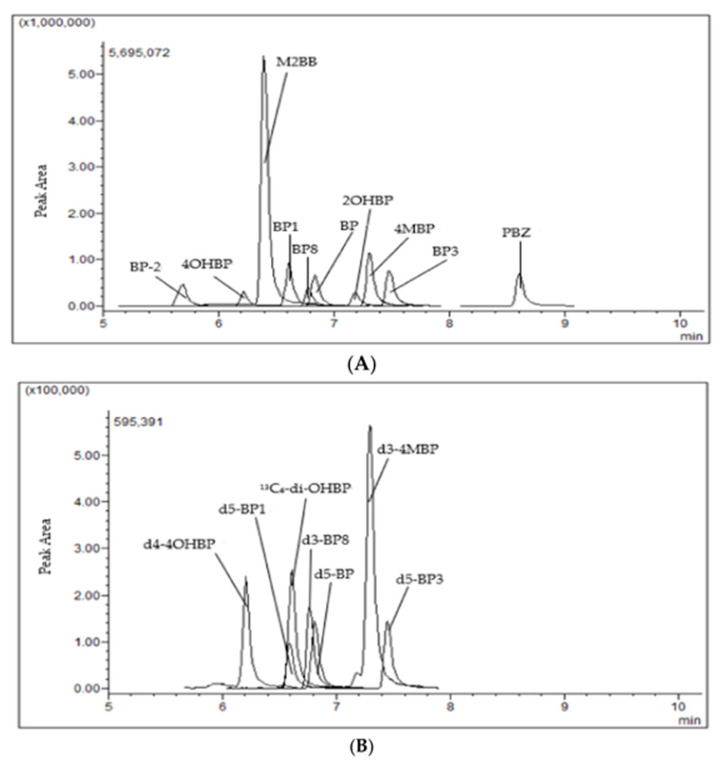
The representative chromatogram of a rice cereal sample spiked with 20 ng/g of the BP and BP analog standards (**A**) and 8 ng/g of the ILIS (**B**).

**Figure 4 foods-11-00572-f004:**
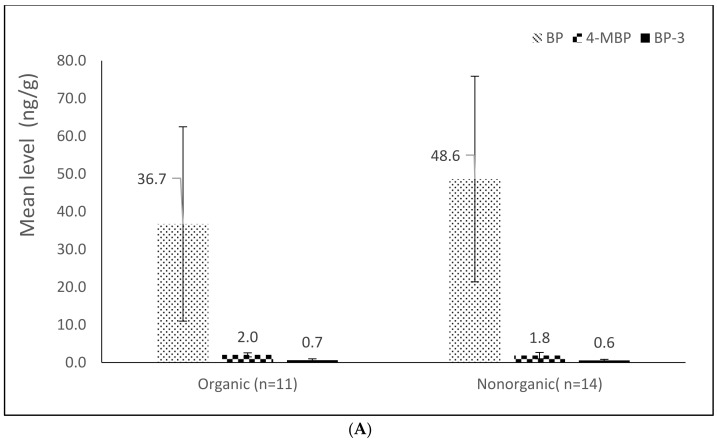
BP, 4-MBP, and BP-3 levels grouped by (**A**) chemical nature and (**B**) packaging materials in rice cereals.

**Table 1 foods-11-00572-t001:** Retention time and MS parameters for quantifying BP and BP analogs and ILISs.

	Mass Spectrometer	Triple Quadrupole Shimadzu Tandem MS (Shimadzu–8045)					
	Ionization	Electrospray Ionization, ESI (Multiple Reaction Monitoring Mode)					
No.	Analytes	Quantification	Qualification	RT (min)	CE 1(v)	CE2 (v)	ESI
1	BP-2	245.0 > 135.0	245.0 > 109.0	5.64	−12	−16	-
2	d_4_-4OHBP	203.1 > 125.2	203.1 > 105.1	6.19	23	23	+
3	4-OHBP	199.0 > 121.1	199.0 > 77.1	6.20	10	10	+
4	M2BB	240.3 > 209.1	240.3 > 152.0	6.37	17	17	+
5	d_5_-BP1	220.1 > 137.0	220.1 > 138.0	6.59	25	25	+
6	BP-1	214.9 > 137.0	214.9 > 105.0	6.61	23	15	+
7	d_3_-BP-8	248.1 > 121.1	248.1 > 154.1	6.77	28	30	+
8	BP-8	245.0 > 121.1	245.0 > 151.0	6.79	13	13	+
9	d5-BP	188.1 > 105.1	188.1 > 110.1	6.80	21	22	+
10	BP	183.0 > 105.1	183.0 > 77.2	6.83	19	19	+
11	^13^C_6_-di-OHBP	221.0 > 137.0	221.0 > 81.0	6.61	11	24	+
12	2-OHBP	199.2 > 121.0	199.2 > 93.0	7.22	11	11	+
13	d_3_-4-MBP	200.2 > 105.1	200.2 > 77.1	7.31	12	12	+
14	4-MBP	197.0 > 105.1	197.0 > 77.1	7.33	21	10	+
15	d_5_-BP_3_	234.0 > 151.0	234.1 > 82.0	7.51	27	26	+
16	BP-3	229.0 > 151.1	229.0 > 105.1	7.54	25	11	+
17	PBZ	259.0 > 105.0	259.0 > 77.1	8.78	10	29	+

**Table 2 foods-11-00572-t002:** Matrix-matched and solvent-matched calibration curves, matrix effect, LOD, LOQ, precision, and accuracy in rice cereal.

Compound	Matrix-Matched Calibration Curve	r	Calibration Curve in Solvent	r	Matrix Effect (%)	LOD (ng/g)	LOQ (ng/g)	RSD (%)		Recovery (%, *n* = 9)
								Intra-Day (*n* = 3)	Inter-Day (*n* = 9)	
BP	y = 0.116x + 0.560	0.999	y = 0.121x + 0.795	0.999	96	0.001	0.003	12.36	16.79	81
BP-1	y = 0.167x − 0.039	0.999	y = 0.175x − 0.095	0.999	96	0.179	0.537	3.22	6.36	82
BP-2	y = 0.093x − 0.068	0.999	y = 0.044x − 0.084	0.998	210	0.015	0.045	4.61	2.60	71
BP-3	y= 0.226x − 0.0626	0.999	y = 0.230x − 0.115	0.999	98	0.061	0.183	1.16	3.18	98
BP-8	y = 0.080x + 0.030	0.998	y = 0.082x + 0.008	0.999	97	0.033	0.099	3.00	8.42	91
2-OHBP	y = 0.011x − 0.017	0.999	y = 0.017x − 0.021	0.999	64	0.317	0.951	5.17	9.56	119
4-OHBP	y = 0.080x + 0.001	0.999	y = 0.083x − 0.008	0.999	96	0.512	1.536	7.82	4.88	89
M2BB	y = 0.716x − 0.281	0.999	y = 0.52x − 0.918	0.998	137	0.391	1.173	4.91	3.79	114
4-MBP	y = 0.135x + 0.007	0.999	y = 0.121x − 0.012	0.999	112	0.009	0.027	3.31	5.64	114
PBZ	y = 0.089x − 0.160	0.999	y = 0.044x − 0.073	0.999	204	0.074	0.222	11.12	15.08	95

RSD: Precision expressed as intra- or inter-day relative standard deviation of spiked 20 ng/g; Recovery: mean recovery (*n* = 9) at final spiked level of 20 ng/g for each analyte.

**Table 3 foods-11-00572-t003:** The BP and BP analog levels in domestic samples of rice cereals grouped by packaging material and chemical nature (*n* = 25) (ng/g).

No.	Packaging Type	Chemical Nature	BP	4-MBP	BP-3	PBZ	4-OHBP
1	Tin Can (*n* = 5)	Organic	25.28	2.05	<LOD	<LOD	<LOD
2		Nonorganic	32.46	<LOD	0.67	<LOD	<LOD
3		Nonorganic	27.39	1.76	0.58	<LOD	<LOD
4		Nonorganic	25.28	2.05	<LOD	<LOD	<LOD
5		Nonorganic	20.51	1.34	0.52	<LOD	<LOD
6	Aluminum Foil Bag (*n* = 12)	Organic	26.00	1.36	0.49	<LOD	<LOD
7		Organic	30.33	1.89	0.54	<LOD	<LOD
8		Organic	34.43	2.12	<LOD	<LOD	<LOD
9		Organic	25.97	1.75	<LOD	<LOD	<LOD
10		Organic	105.16	1.99	0.73	<LOD	<LOD
11		Organic	53.88	2.42	<LOD	<LOD	<LOD
12		Nonorganic	108.93	<LOD	<LOD	<LOD	<LOD
13		Nonorganic	28.49	1.55	0.47	<LOD	13.98
14		Nonorganic	68.79	2.36	0.68	<LOD	<LOD
15		Nonorganic	19.52	1.57	0.44	<LOD	33.26
16		Nonorganic	42.44	2.21	<LOD	<LOD	<LOD
17		Nonorganic	43.98	2.22	<LOD	<LOD	<LOD
18	Plastic Bag (*n* = 8)	Organic	32.33	2.09	0.58	<LOD	<LOD
19		Organic	19.13	1.80	0.69	<LOD	<LOD
20		Organic	33.33	1.37	0.69	<LOD	<LOD
21		Organic	70.79	3.31	<LOD	<LOD	<LOD
22		Nonorganic	90.53	<LOD	<LOD	0.94	<LOD
23		Nonorganic	50.99	1.61	<LOD	<LOD	<LOD
24		Nonorganic	63.59	1.91	<LOD	<LOD	<LOD
25		Nonorganic	64.99	1.87	<LOD	<LOD	<LOD
	Compounds		BP	4-MBP	BP-3	PBZ	4-OHBP
	Detection frequency (%)		100	88	52	4	8
	Mean (SD)		45.8 (26.3)	1.9 (0.4)	0.6 (0.1)	-	23.6 (13.6)
	Geometric Mean		39.8	1.9	0.6	-	21.6
	Minimum–Maximum		19.13–108.93	<LOD–3.3	<LOD–0.7	-	<LOD–33.3

## Data Availability

All data and its Appendix A files generated or analyzed during this study are included in this published article.

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
