# Peer review of "Identification of Benzophenone Analogs in Rice Cereal through Fast Pesticide Extraction and Ultrahigh-Performance Liquid Chromatography–Tandem Mass Spectrometry"

_foods, 2022, doi:10.3390/foods11040572_

Round 1

Reviewer 1 Report

The topic is interesting and covers the Aims & Scope of the Journal. However significant amendments shall be performed. 

GENERAL COMMENTS

My general comments is that from the discussion part is missing the discussion regarding the comparison of this method in comparison to the existing LC/UHPLC-MS methodologies or even more importantly with the GC-MS methodologies. it shall not be omitted that GC-MS is a method of choice for the analysis of these analyses,

SPECIFIC COMMENTS

Abstract/L.17-18: Please change to “precision” results and not intraday and interlay. Please include the recoveries results.

l.47: The EU has regulated specific limits and not set! “I think the term regulated” fits better.

BPs:BPs occur also as PI (photoinitiators). Please  consult :
Tsochatzis et al. 2020 ABC https://link.springer.com/article/10.1007%2Fs00216-020-02758-7
Amend references, accordingly.

L.52-54: In this part it shall be connection with the FCM. Significant  contamination might occur from FCM.

L.55-69:Benzophenones are analyzed also with GC-MS. The authors shall update the Introduction and discussion in respect to the use of GC-MS for their analysis.

Paragraph 2.1: please provide the full chemical names along with the abbreviations used.

Paragraph 2.4: However no description of the conditions and parameters exist in the experimental part. Please add.

Paragraph 2.4: According to Table 1, the authors use both ESI negative and positive mode. This part shall be described and explained in this paragraph.

Figure 2: In my opinion the selection of colors of this Figure shall be improved. Please revise in order to help the readers understand better this figure.

L.177-193 are referring to experimental conditions of the uHPLC-QQQ-MS system. Why they are presented in the Results section? They shall be placed in the experimentla section.

Paragraph 3.2: The authors shall describe the switch from ESI(-) to ESI(+) during run.

Figure 3: In my opinion, Figure 3 is too long and shall be shortened. Also the X- and Y- axis font is too small.

Paragraph 3.3: Please associate whenever is necessary "Recovery" with trueness (Accuracy) and the RSDs with for precision (Intra-, inter-day).

Paragraph 3.4 and Table 3: Why the authors use the term “organicity”? 
Maybe chemical nature is a more appropriate term to use.Please revise and amend. Please improve discussion in regard to “organicity”.

Figure A1 shall be placed to a supplementary file. 

Author Response

Dear Dr. Reviewer,

Reviewer 2 Report

In the paper Identification of Benzophenone Analogs in Rice Cereal through Fast Pesticide Extraction and Ultrahigh-Performance Liquid Chromatography–Tandem Mass Spectrometry , authors report a a fast, robust, and sensitive analytical method for the simultaneous identification of benzophenone (BP) and nine BP analog.
Paper is good write and support by results. 
However, several aspect can be emproved. 
I suggest major revision. 

Introduction line 58. 
Please add tha as recently reported in literature, pollutant extraction is  a critical point in chemical analyses. 
In this context add several refferences such as Study on the Stability of Antibiotics, Pesticides and Drugs in Water by Using a Straightforward Procedure Applying HPLC-Mass Spectrometric Determination for Analytical Purposes. Separations, 2021, 8.10: 179.

Moreover in the sentence 
Diet is a major source of BPs exposure in the general population. Although BP levels in foodstuffs are too low to constitute a risk, long-term and cumulative exposure to BPs may have adverse effects. Thus, a convenient and reliable analytical method for quantifying BP contents in foods is required. Solvent extraction with acetonitrile [15,16] and pres- surized liquid extraction (PLE) [17] have been proposed for determining the level of BPs in breakfast cereals; however, these techniques are time-consuming and require a large volume of organic solvents. Solvent extraction followed by solid-phase extraction (SPE) 
with a hydrophilic–lipophilic balance (HLB) or silica cartridge [18,19];

Pleas add othe referfences to underline the use of cardrige for other pollutants such as hormones. 
In this contex add 
Determination of estrogenic endocrine disruptors in water at sub-ng L− 1 levels in compliance with Decision 2015/495/EU using offline-online solid phase extraction concentration coupled with high performance liquid chromatography-tandem mass spectrometry. Microchemical Journal, 2019, 147: 1186-1191.

Table 1 quantification
Please report only one significative cifre for transitions

4. Conclusions
Please add smore conclusions

Author Response

Dear Dr. Reviewer,

Round 2

Reviewer 2 Report

All corrections were made. 

Paper can be accept